# Boosting Speech Recognition Robustness to Modality-Distortion with Contrast-Augmented Prompts

## ABSTRACT

In the burgeoning field of Audio-Visual Speech Recognition (AVSR), extant research has predominantly concentrated on the training paradigms tailored for high-quality resources. However, owing to the challenges inherent in real-world data collection, audio-visual data are frequently affected by modality-distortion, which encompasses audio-visual asynchrony, video noise and audio noise. The recognition accuracy of existing AVSR method is significantly compromised when multiple modality-distortion coexist in low-resource data. In light of the above challenges, we propose PCD: *cluster-Prompt with Contrastive Decomposition*, a robust framework for modality-distortion speech recognition, specifically devised to transpose the pre-trained knowledge from high-resource domain to the targeted domain by leveraging contrast-augmented prompts. In contrast to previous studies, we take into consideration the possibility of various types of distortion in both the audio and visual modalities. Concretely, we design bespoke prompts to delineate each modality-distortion, guiding the model to achieve speech recognition applicable to various distortion scenarios with quite few learnable parameters. To materialize the prompt mechanism, we employ multiple cluster-based strategies that better suits the pre-trained audio-visual model. Additionally, we design a contrastive decomposition mechanism to restrict the explicit relationships among various modality conditions, given their shared task knowledge and disparate modality priors. Extensive results on LRS2 dataset demonstrate that PCD achieves state-of-the-art performance for audio-visual speech recognition under the constraints of distorted resources.

## CCS CONCEPTS

• **Computing methodologies** → **Speech recognition**; *Computer vision tasks.*

## KEYWORDS

multi-modal learning, audio-visual speech recognition, modality-distortion

## 1 INTRODUCTION

Audio-Visual Speech Recognition (AVSR), which leverages the synergistic interaction between human speech and temporally aligned lip movement videos to generate natural language, has emerged

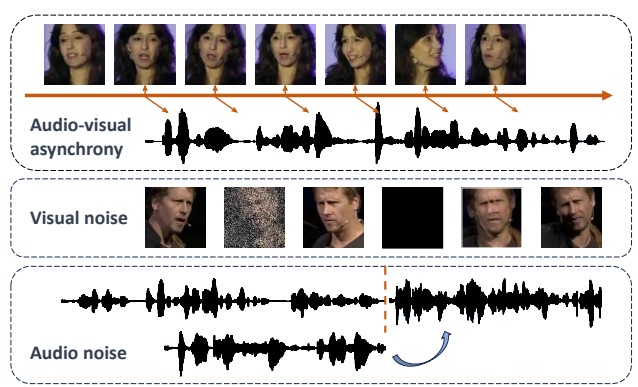

**Figure 1: Examples of potential distortion in audio-visual datasets. Audio-visual asynchrony: the audio and video are out of sync temporally or mismatched across different segments; Visual noise: the video is disturbed by blackouts, blurriness, frame drops, or screen flickering; Audio noise: the audio contains environmental noise.**

as a vibrant frontier in the applications of multi-modal learning [1, 19, 20, 27]. With learning audio-visual features, AVSR has demonstrated superior performance compared to single-modality input models (including audio-only and visual-only input models). While the complementarity of audio and visual modalities is often assumed, the reality frequently deviates from this ideal scenario due to recording equipment and environmental constraints. Instances where data is distorted, such as audio-visual asynchrony induced by storage device malfunctions, audio noise in outdoor interview scenes, or visual noise in video conferencing scenarios (as shown in Fig. 1), often result in the confusion of AVSR model. With modality-distortion, models may even fall short in effectiveness compared to their single-modal counterparts, especially given the presence of multiple distortion scenarios. On the other hand, distorted data can possess inherent value due to the challenges associated with acquiring high-resource datasets and the high cost of label annotation, particularly evident in endeavors such as the preservation of minority languages or data recovery. The ubiquitous presence of distortion in real-life scenarios poses substantial challenges to the application of AVSR.

Established AVSR models are typically trained on high-resource datasets, with the objective of attaining peak performance under circumstances of data completeness or in scenarios where one modality exhibits incompleteness. For instance, numerous AVSR investigations [2, 19, 28, 36] evaluate the efficacy of models amidst audio noise, thereby validating methodological robustness. Concurrently, alternative research endeavors [4, 5, 10] concentrate on addressing issues pertaining to visual modality missing or noisy.

However, none of these methods address the challenge of training with low-quality datasets containing simultaneous audio and visual modality-distortion which are commonly encountered in real-life situations, lacking robustness across diverse distortion scenarios. Training a separate optimal model for each potential distortion scenarios is ideal but impractical, given the substantial computational resources required. Constrained by the availability of high-quality datasets, there is a need for more robust approaches that consume fewer computational resources to adapt to data with modality-distortion.

Incorporating three real-world concerns into consideration: (i) distortion occurrences are always diverse and scattered across various modalities; (ii) distortion often stems from equipment and environment limitations, hence the distortion forms within the same batch of data may be unique and uniform, limited to a single modality; (iii) in scenarios where both modalities suffer from severe distortion, the data is chaotic and devoid of value, leading to confusion in model even human recognition. Thereby, we introduce a general setting to simulate real-world modality-distortion, where the dataset comprises three types of scenarios: clean data, data with audio distortion, and data with visual distortion. We propose a novel method called PCD: Cluster-Prompt with Contrastive Decomposition to enhance speech recognition robustness across diverse modality-distortion conditions within a unified framework. Drawing inspiration from the notable success of prompt learning within the field of multi-task fine-tuning in natural language processing [8, 18, 37], we design tailored prompts for each modality-distortion conditions instead of training individual model with a myriad of parameters. Building upon a fully pre-trained transformer-based model with audio-visual alignment [27], we effectively utilize prompts to facilitate the transfer of knowledge from a pre-trained high resource domain to a low-quality domain with modality-distortion. In order to enhance compatibility with pre-trained models, we develop a cluster module and explore two attachment strategies. Following the computation of features by the cluster module, generated prompts are then combined with either the input or key&value during multi-head self-attention operations. Furthermore, we employ low-rank decomposition and contrastive regularization term, supervising the task-specific part of cluster-prompts to provide more refined guidance tailored to particular scenarios. Under the explicit constraint of task interaction, the modality-distortion tasks prompts tends to approach the clean task prompts while diverging from each other, allowing prompts to learn more task-relevant features. The main contributions are as follows:

- We propose a novel framework, PCD, which is the first work dedicated to enhancing robustness in modality-distortion speech recognition.
- We introduce two cluster-based strategies tailored for implementing the prompt mechanism, which are especially optimized to complement pre-trained audio-visual models.
- We design a novel contrastive decomposition mechanism for prompts, aiming to mine the interactions between diverse modality-distortion conditions.
- PCD achieves the state-of-the-art performance on the LRS2 dataset, demonstrating its outstanding efficacy in AVSR tasks involving modality-distortion.

## 2 RELATED WORK

### 2.1 Audio-Visual Speech Recognition

Recently, AVSR which aims to translates synchronized audio and video into corresponding text, has been attracting increasing research interest as it presents a viable solution for employing the fusion of audio and visual modalities as an alternative to ASR[21, 23, 26]. TM-seq2seq [1] first introduce transformer architecture into AVSR task, utilizing pre-computed visual features and audio Log-Mel filter features as inputs. E2E Conformer [20] leverages Conformer architectures [7] to extract visual and audio features, facilitating end-to-end training. Moreover, LUSSL-AVSR [22] utilize self-supervised learning for AVSR task by incorporating the pre-trained model trained in massive unlabelled single modality data. Similarly employing self-supervised learning, AV-HuBERT [27] learns the correspondence of audio and video modalities by masking multi-stream video input and predicts automatically discovered and iteratively refined multi-modal hidden units. Recently, Auto-AVSR [19] effectively expand the audio-visual dataset by utilizing pre-trained ASR models to automatically transcribe unlabeled video data.

Typically, AVSR research leverages the visual modality to enhance robustness against audio noise [16, 17, 28, 34], while some studies also address potential video noise in audio-visual dataset. [3, 4] tackles scenarios with missing video frames, while [9] focuses on resolving occlusions that may occur in videos. However, the aforementioned methodologies only address a singular type of distortion, thus lacking robustness across diverse scenarios. In contrast, this paper conducts a more thorough study on AVSR's robustness where various modality-distortion would occur for any data sample and anywhere in learning phases, particularly focusing on reducing the computation of model fine-tuning.

### 2.2 Prompt Learning

In prompt-driven approaches, task-specific textual descriptions or cues are utilized to guide models towards integrating and concurrently processing data originating from various sensors, sources, or formats such as text, images, audio, or video [6, 14, 18]. This methodology has found extensive usage within the field of natural language processing and has recently been introduced into vision problems [12, 33, 38], audio generation [11, 32] and multi-modal learning tasks[35, 40]. [15, 25] introduce prefix tuning, exploring additional interactions between prompts and pre-trained model. [14, 31, 39] fine-tune pre-trained models by optimizing continuous set of prompt vectors called soft prompt instead of hand-crafted prompts. In multi-modal tasks, MaPLe [13] applies prompts in both vision and language encoders to improve the alignment between vision and language representation. TRIPLET [24] further employs decoupled prompts and prompt interaction strategies to capture the complex interactions between modalities. These studies investigate the remarkable adaptability of prompt learning across various tasks involving diverse input domains. Inspired by the aforementioned work, we introduce prompt learning into AVSR task, transferring knowledge from high-resource domains to target domains containing various types of modality-distortion which can be regarded as different learning tasks. We further experiment with various prompt strategies to better align with the pre-trained AVSR model.

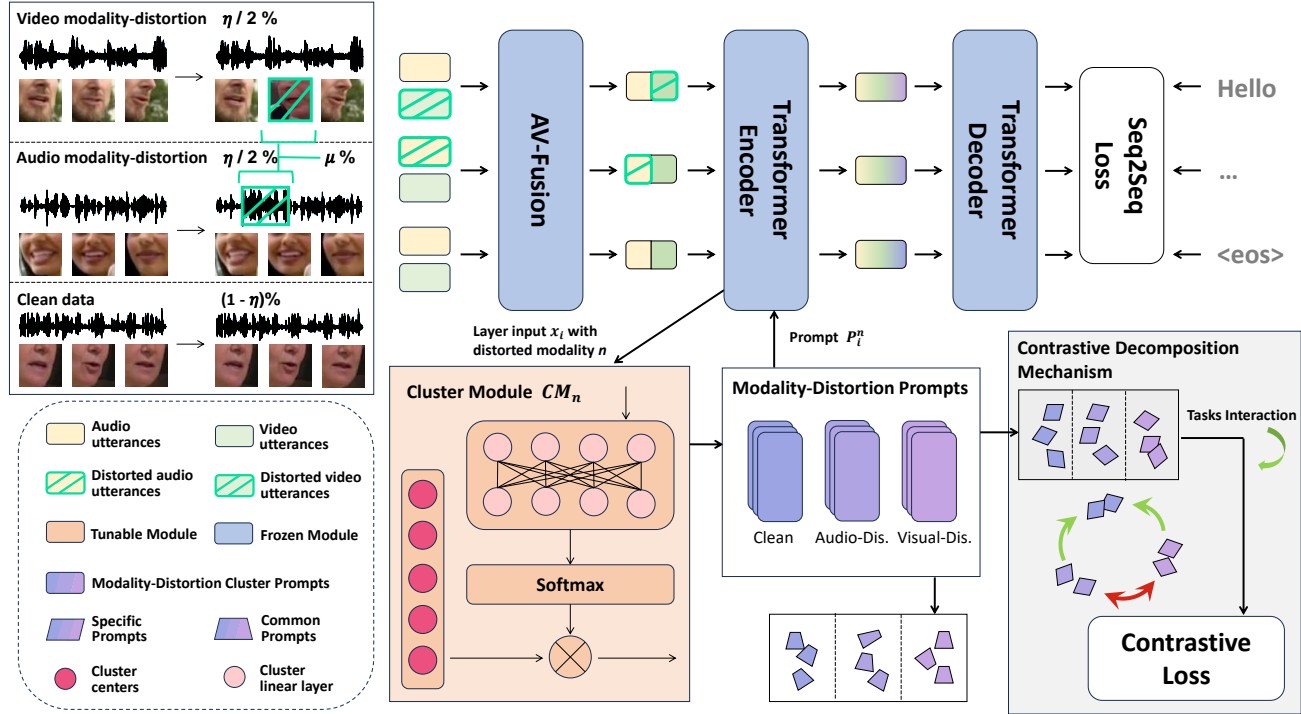

**Figure 2: The overall framework of our proposed PCD approach. Upon pre-training a audio-visual alignment transformer on high-resource dataset, we freeze the structure and finetune the prompts on distorted dataset. Specifically, we train the cluster module $CM_n$ tailored to data with distorted modality $n$ to generate bespoke prompts. In addition to the seq2seq loss, a contrastive decomposition mechanism is utilized to supervise the learning of task-specific features in prompts.**

## 3 METHOD

### 3.1 Problem Formulation

We formulate the problem setting for AVSR with modality-distortion in this section. Suppose we have an audio-visual dataset contains modality-distortion $\tilde{D} = \{\tilde{A}, \tilde{V}, S\}$, where $\tilde{A}, \tilde{V}$ represents the audio and video utterance that may be contaminated by random distortion, and $S$ represents corresponding natural sentence. Neglecting the scenario where distortion concurrently affects both modalities, as explained earlier, we partition the dataset into three subsets: clean data $D_c = \{A, V, S\}$, data with audio distortion $D_{ad} = \{\tilde{A}, V, S\}$, data with video distortion $D_{vd} = \{A, \tilde{V}, S\}$. Under such real-world conditions, there are two challenging problems, one is to adapt one model framework to multiple types of distortion while minimizing computational resources. The other is to avoid confusion from distorted data during the training process.

### 3.2 Transformer with Audio-Visual Alignment

Since AVSR can be viewed as a sequence-to-sequence transformation task, current state-of-the-art AVSR methods are all based on transformer structures. To fully exploit multi-modal knowledge, we employ the audio-visual aligned encoder, similar to AV-Hubert[27], which is a self-supervised representation learning method for audio-visual speech. The AV-Hubert structure integrates and extracts

audio-visual features from raw data, which are then utilized by a transformer decoder to generate natural sentences.

The pre-training process of AV-Hubert alternates between feature clustering and mask prediction. The model leverages clustering to generate self-supervised targets and strengthens cross-modal fusion through mask prediction, facilitating the mapping of audio and video sequences into a unified phoneme space $f^p \in \mathbb{R}^{T \times D}$ where $T$ is the length of the sequence and $D$ is the dimension of the embedding.

Upon acquiring audio-visual representations through self supervised methods, the seq2seq loss is utilized to train the entire model, including the decoder, and also serves as part of the objective for prompt training:

$$\mathcal{L}_{s2s} = - \sum_{t=1}^{s} \log p(w_t \mid \{w_i\}_{i=1}^{t-1}, f^p) \qquad (1)$$

where $\{w_i\}_{i=1}^{s}$ is the ground-truth transcription.

Due to its superior performance on both multi-modal and uni-modal tasks, we choose the AV-Hubert as our backbone model, pre-trained on large-scale vision and audio datasets. Amidst encountering data distortion in one modality, the exceptional performance of AV-Hubert in uni-modality speech recognition facilitates a more effective guidance to prioritize the clean modality. However, the cost of training a full AV-Hubert model to a specific distorted

condition is prohibitive, and practical tasks often involve diverse types of modality-distortion that cannot be addressed by a single model. So we design prompts tailored to different combinations of data distorted on a pre-trained AV-Hubert model to transfer knowledge from a high-resource domain to a low-resource domain with minimal training cost.

## 3.3 Cluster-Prompt for Modality-Distortion

Following pre-training, the subsequent step involves guiding the model to acclimate to distorted data, which often exhibits various types of distortion and is characterized by limited quantity in real-world scenarios. To guide the model pay more attention to the clean part in distortion contaminated audio-visual pairs, we design bespoke prompts for each conditions which are collections of trainable vectors, interacting with the model. Typically, for the tasks set $N$, we assign $|N|$ kinds of prompts where the number is three for training setting simulating real-world modality-distortion scenarios, as formulated in Section 3.1. The corresponding prompts are concatenated to designated positions of the multi-head self-attention (MSA) module.

In the preceding step, we adopt AV-Fusion to map audio and visual representations into the same phoneme space to enable the model to attain recognition capabilities across uni-modality, making it more adept at handling modality-distortion. Initially, the concatenation of audio and visual utterances $u^{av} = \text{concat}(u^a, u^v) \in \mathbb{R}^{T \times 2D}$ is fed into AV-Fusion to obtain the fused features $f^m \in \mathbb{R}^{T \times D}$. Using $f^m$ as the input to the first layer of the transformer encoder, we denote the input fused features of the $i$-th MSA layer as $x_i \in \mathbb{R}^{T \times D}, i = 1, 2, \ldots, M$ with number of layers $M$. Based on the input distortion type, we choose the respective prompts $p_i^n \in \mathbb{R}^{L_p \times D}$ with prompt length $L_p$ and representations for different modality-distortion cases $n \in N \equiv \{c, ad, vd\}$, which are then interacted with $x_i$ to generate extended features $x_i^p$:

$$x_i^p = F_{\text{prompt}}(p_i^n, x_i) \tag{2}$$

where $F_{\text{prompt}}$ defines the attach approaches for prompts to interact with the designated structures in MSA layers.

### 3.3.1 Cluster-Based Prompts.
In order to extract valuable features from distorted data, we adopt a cluster strategy for prompt generation. In the pre-training process, a k-means approach is employed to extract cluster labels on audio-visual features. Building upon this concept, we employ an tunable cluster module $CM_n$ to cluster the inputs with modality-distortion condition $n$, thereby generating corresponding prompts. For audio-visual features that exhibits greater similarity, cluster module facilitates prompts in offering more proximate guidance to the pre-trained model. Specifically, the input $x_i$ is first fed into a extraction network, which is consisted of a linear projection layer, a summation operation and a cluster-wise softmax layer to extract the cluster weights for each phoneme features. We define cluster centers $c_i \in \mathbb{R}^{N_c \times L_p \times D}$ in each layer $i$ of the encoder, where $N_c$ is the number of the clusters and compute prompts by combining them with the clustering results of the input:

$$p_i = \text{Extract}(x_i) \times c_i \tag{3}$$

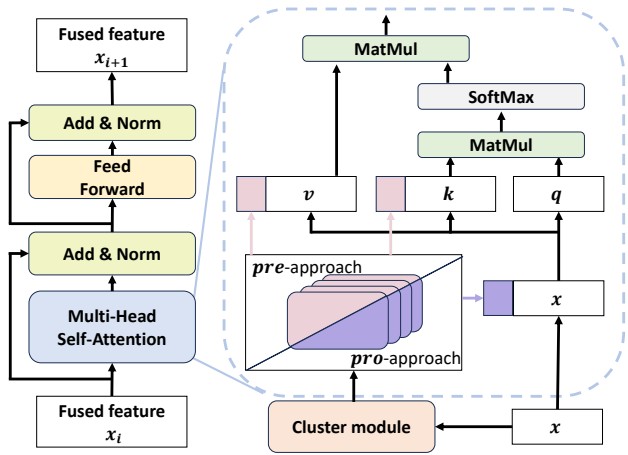

**Figure 3: The illustration of two prompts attach strategies. After being generated by the cluster module, prompts can be concatenated with the input (*pro*-approach) within the multi-head self-attention (MSA) block, or alternatively, they can be concatenated with the key&value pairs (*pre*-approach).**

In the situation of modality-distortion, cluster module strengthens the connections between similar clusters of phoneme features, learning features about the distorted segment.

### 3.3.2 Prompt Attach Strategies.
The design of the $F_{\text{prompt}}$ function, as outlined in Equation 2, is crucial for integrating prompts with pre-trained models to transfer knowledge from a high-quality domain to a target low-quality domain. We adopt two interaction mechanisms with the MSA module, as shown in Fig. 3.

We denote the query, key and value in the $i$-th MSA layer as $Q_i, K_i, V_i$, which are obtained by applying a projection matrix to $x_i$. The first strategy which is inspired by the concept of soft prompt tuning ($Pro$) is to prepend prompts with input sequences for each layer, which is equivalent to concatenate the same prompt parameters to $Q_i, K_i$ and $V_i$. The prompt function can be written as:

$$F_{\text{prompt}}^{\text{Pro}}(p_i^n, x_i) = \text{ATTEN}_i([p_i^n; Q_i], [p_i^n; K_i], [p_i^n; V_i]) \tag{4}$$

where $[\ldots; \ldots]$ represents the concatenation operation. With the implementation of $Pro$-approach prompts, attention mechanisms are more targeted towards feature processing, and each layer's input token $x_i$ contains inherited prompt information from the previous layers, leading to more effective instructions for the model prediction.

Another prompting approach, inspired by prefix tuning ($Pre$), focuses on the key and value at the MSA layer. We split the prompt $p_i^n$ into two sub-prompts $p_i^k$, $p_i^v$ and prepend them to the key and value vectors respectively. We can define the prompt function for $Pre$-approach prompts as:

$$F_{\text{prompt}}^{\text{Pre}}(p_i^n, x_i) = \text{ATTEN}_i(Q_i, [p_i^k; K_i], [p_i^v; V_i]) \tag{5}$$

The attention-level prompting provides another way to instruct the pretrained model from the perspective of the attention mechanism in transformers.

 

## 3.4 Contrastive Decomposition

To supervise prompt learning for specific modal combinations with distortion, we employ a common-specific decomposition approach. Specifically, we employ a low-rank decomposition mechanism to map the information from $p^n$ into the common represents $p_c \in \mathbb{R}^{r \times d}$ and task-specific represents $p_s^n \in \mathbb{R}^{L_p \times r}$ where $r$ donates the rank of the matrix decomposition, which can be formulated as:

$$p^n = p_s^n \cdot p_c \tag{6}$$

where $\cdot$ denotes matrix multiplication. This was done to distinguish between the common features of the AVSR task and the specific characteristics of a particular modality-distortion condition. Rethinking the cluster process, the low-rank decomposition of prompts $p_i^n$ is equivalent to the same operation applied to cluster centers $c_i^n$ when implementing.

To constrain the implicit interaction between prompts, we focus on the explicit connections between tasks. We anticipate similar tasks to entail prompts that provide more analogous guidance to the model. In the context of modality-distortion in AVSR, our objective is to attain model performance comparable to clean data even in the presence of audio and video distortion, which means narrowing the gap between prompts in the clean domain $p_s^c$ and those in the distorted domains $p_s^{ad}, p_s^{vd}$ while widening the separation between the latter two. Specifically, we propose a contrastive loss following InfoNCE Loss[30] with specific prompt $p_s^n$ to supervise common-specific decomposition contrastive learning:

$$\mathcal{L}_{cl} = \sum_{n \in N} -\frac{1}{|C(n)|} \sum_{m \in C(n)} \log \frac{\exp \mathrm{sim}\left(p_s^n, p_s^m\right)/\tau}{\sum_{k \in N \setminus \{n\}} \exp \mathrm{sim}\left(p_s^n, p_s^k\right)/\tau} \tag{7}$$

where $C(n)$ represents the set of tasks that have a closer relationship to task $n$ (e.g. when $n$ refers to the task dealing with data with audio distortion, $C(n)$ contains the task with clean data), $|C(n)|$ is its cardinality, $\tau$ is the temperature parameter, and $sim(a, b)$ denotes the similarity between vectors $a$ and $b$. Drawing from an analysis of explicit task relationships, positive and negative sample sets are derived. Contrastive learning is then applied to encourage similar tasks to learn more similar prompts, facilitating the extraction of task-related information and enhancing the effectiveness of guiding the model.

For the overall objective of the prompt training, we apply the $\mathcal{L}_{cl}$ and cross-entropy loss $\mathcal{L}_{s2s}$ in Eqn. 1 with the scale factor $\alpha$:

$$\mathcal{L}_{overall} = (1 - \alpha)\mathcal{L}_{s2s} + \alpha\mathcal{L}_{cl} \tag{8}$$

## 4 EXPERIMENT

### 4.1 Dataset

*LRS2.* [1] stands out as one of the most widely utilized publicly accessible English lip-reading datasets, including 224 hours of video content sourced from BBC television programs. This dataset originally comprises two partitions for training: Pretrain (195h) and Train (29h), both transcribed at the sentence level from video to text. The key disparity lies in the fact that video clips in the Pretrain partition are not rigorously trimmed and may exceed the corresponding text length. Our experiments on LRS2 involving varying training

data amounts, specifically comparing Pretrain+Train (224h) against Train (29h).

*LRS2-DISTORTED.* Based on the LRS2 dataset, we further propose the LRS2-DISTORTED to verify the robustness to the modality-distortion speech recognition with low-resource training data. We introduce various types of modality-distortion into the LRS2 dataset, aiming to simulate realistic scenarios where audio and visual distortion randomly occurs across both training and testing phases. Note that to ensure fair training, the specific distortion data replaced or added for each sample is predetermined. Meanwhile, both the distortion rate and the distortion types can be varied to compare the robustness of the models.

### 4.2 Metric

For all experiments we use the word error rate (WER) as the evaluation index of speech recognition. WER can be defined a $WER = (S + D + I)/M$, where $S, D, I, M$ represent the number of words replaced, deleted, inserted and referenced respectively.

### 4.3 Implementation Details

*4.3.1* **Modality-Distortion Setting**. We focus on a more practical scenario where distortion is prevalent both in training and testing phases. We define distortion rate $\eta$ as the proportion of modality-distortion data to the entire dataset, and $\mu$ as the proportion of distortion present in each individual sample. Scenario with distortion rate $\eta$ and $\mu$ indicates that there are $\eta/2$ data with audio distortion, $\eta/2$ data with video distortion, and $(1 - \eta)$ complete data, where $\mu$ of each sample is replaced by data with distortion. To validate the robustness of the method, we employ three approaches to simulate distortion. Approach $a$ simulates a severe scenario entailing replacing segments of either the audio or video with segments from another sample, and Approach $b$ involved adding MUSAN [29] noise to the audio and introducing screen flickering to the video. Approach $c$ represents temporal asynchrony between audio and video, a prevalent type of distortion in real-world scenarios. Specifically, it entails delaying the data of the distortion modality by a specified number of frames. In ablation experiments, we default to setting $\eta = 70\%, \mu = 80\%$ for inference with condition $a$.

*4.3.2* **Experimental Details** . The model is trained on NVIDIA GeForce RTX 3080Ti GPU, equipped with 10GB of VRAM. Constrained by computational resources and the simulation of low-resource data, we conduct ablation experiments mainly on the base transformer model on the LRS2-29h dataset.

### 4.4 Main Result

In this section, we compare the performance of our approach with the backbone and other AVSR methods under modality-distortion setting on LRS2, as presented in Table 1. Limited by data distortion, the performance of these baselines deviates significantly from the result training with clean data, indicating the lack of robustness to modality-distortion. Comparatively, our proposed PCD outperforms the state-of-the-art AVSR across various distortion settings, achieving significant reductions in the metric (up to 3% on WER) compared to the backbone, AV-Hubert. It is noteworthy that all improvements are derived from approximately 2% of the parameters

**Table 1: The WER (%) performance on LRS2-DISTORTED under the modality-distortion AVSR setting under different evaluating distortion rates with distortion approach _a_. Models are trained under $\eta = 70\%$, $\mu = 60\%$.**

| model | | training set | clean | $\eta=50\%$ | | $\eta=70\%$ | | $\eta=90\%$ | |
|---|---|---|---|---|---|---|---|---|---|
| | | | | $\mu=60\%$ | $\mu=80\%$ | $\mu=60\%$ | $\mu=80\%$ | $\mu=60\%$ | $\mu=80\%$ |
| TM-Seq2seq [1] | | | 12.20 | 25.26 | 28.63 | 32.85 | 37.44 | 39.71 | 46.51 |
| End2end Conformer [20] | | 224h-distorted | 6.13 | 13.12 | 15.97 | 14.92 | 19.89 | 19.78 | 24.32 |
| LUSSL-AVSR [22] | | | 4.95 | 10.73 | 12.76 | 13.98 | 16.53 | 17.01 | 21.29 |
| Auto-AVSR [19] | | | 4.01 | 10.41 | 11.98 | 12.63 | 15.20 | 16.31 | 19.23 |
| transformer-base | AV-HuBERT | 29h-distorted | 6.81 | 14.22 | 16.71 | 16.92 | 20.20 | 20.18 | 24.47 |
| | $PCD_{pre}$ | | **5.56** | **11.79** | 14.70 | **14.65** | 18.90 | 17.28 | 22.91 |
| | $PCD_{pro}$ | | 5.73 | 11.82 | **14.51** | 14.79 | **18.25** | **17.12** | **22.12** |
| | AV-HuBERT | 224h-distorted | 4.36 | 10.56 | 12.60 | 12.83 | 15.83 | 17.04 | 19.43 |
| | $PCD_{pre}$ | | **3.89** | **9.32** | 11.43 | 11.52 | 14.79 | 15.89 | 18.11 |
| | $PCD_{pro}$ | | 3.95 | 9.61 | **11.38** | **11.47** | **14.41** | **15.65** | **17.93** |
| transformer-large | AV-HuBERT | 29h-distorted | 5.69 | 11.29 | 14.45 | 13.71 | 18.40 | 16.13 | 22.18 |
| | $PCD_{pre}$ | | **4.78** | **9.51** | 13.12 | 12.31 | 16.80 | 14.21 | 20.48 |
| | $PCD_{pro}$ | | 4.80 | 9.81 | **12.54** | **12.19** | **16.44** | **13.89** | **19.56** |
| | AV-HuBERT | 224h-distorted | 3.54 | 8.13 | 10.71 | 10.23 | 14.20 | 11.77 | 16.44 |
| | $PCD_{pre}$ | | **3.28** | **7.04** | 9.82 | 9.13 | 13.51 | 9.95 | 14.78 |
| | $PCD_{pro}$ | | 3.32 | 7.23 | **9.53** | **8.99** | **13.12** | **9.83** | **14.21** |

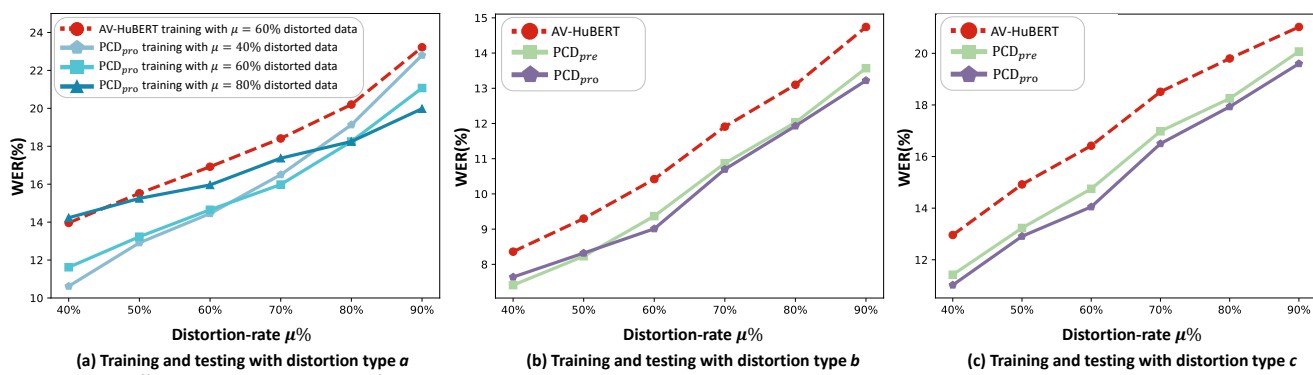

(a) Training and testing with distortion type _a_ and different training distortion-rate $\mu\%$

(b) Training and testing with distortion type _b_

(c) Training and testing with distortion type _c_

**Figure 4: Performance of PCD under different training distortion rates and its behavior under various distortion conditions.**

of AV-Hubert. The poor performance of the baselines is attributed to their focus on utilizing complete modalities, leading inadequate adaptation to modality-distortion scenarios. In contrast, benefiting from the prompts designed for various distortion conditions, PCD learns how to leverage pre-trained comprehensive knowledge to tackle different situation. The cluster-prompt module offers a robust instructional framework for guiding model predictions. Moreover, the contrastive decomposition constraint enhances the interaction between prompts, while learning task-specific features strengthens the robustness to distortion settings. The main results convincingly illustrate the effectiveness of our proposed method.

From the perspective of distortion settings, as the distortion rate $\eta$ increases, the magnitude of improvement consistently rises, indicating that PCD's guidance on distorted samples is stronger than on clean data. PCD also demonstrates greater robustness to varying distortion rates $\mu$, with a corresponding increase in improvement.

From the perspective of model strategy analysis, the _pre_-approach demonstrates superior performance under low-distortion settings, whereas the ability of _pro_-approach to convey information across different layers renders it more suitable for high-distortion data. From a data quantity perspective, it is observed that PCD exhibits a more significant enhancement on the 29h dataset than 224h since sufficient data has enabled the backbone to acquire more knowledge and adapt to the distortion settings. In other words, PCD not only ensures stable improvements over 224h dataset but also demonstrates greater suitability for low-quality target domains, which aligns with the primary application scenario proposed.

### 4.5 Ablation study

_4.5.1_ **Robustness to different distortion setting.** In the main result, we validate the robustness of the PCD method to varying distortion rates during the testing phase with distortion type _a_.

In this section, we conduct additional experiments to explore the performance of PCD under different training distortion rates and its behavior under various distortion conditions. In Figure 4 (a), we test the performance of models trained under different distortion rates $\mu$. Typically, using data with lower distortion rates allows the model to acquire more knowledge, as evident from the test results under a 40% distortion rate. However, models trained with too little distortion are not adept at tasks with excessively high distortion rates. Moderate levels of distortion can aid in improving the model's generalization ability. Based on this, we select the model trained under the 60% distortion condition, which exhibit the optimal trade-off metric. In Figure 4 (b, c), while keeping other settings constant, we conduct a comparison between PCD and the baseline under the additional distortion settings $b, c$ mentioned in section 4.3.1. It can be observed that the PCD method exhibits improvements when confronted with different real-world distortion scenarios and robustness across varying distortion rates.

*4.5.2* **The Impact of cluster center.** The Cluster Prompt module implicitly learns distinct features of the modality-distortion by distinguishing between different phoneme features, enabling the model to provide targeted prompts. In Figure 5, we show the influence of the number of clusters on the performance of accuracy. In contrast to abstaining from the cluster strategy (i.e., with zero cluster centers), the cluster module affords the model a degree of flexibility in addressing the modality-distortion, consequently enhancing recognition accuracy. The optimal performance under the 29h dataset is achieved with 30 cluster centers, whereas the number is 80 for the 224h dataset. This indicates that a higher number of cluster centers represent a finer handling of distorted data. But excessive cluster centers result in insufficient data to adequately train these parameters, leading to a decline in recognition performance.

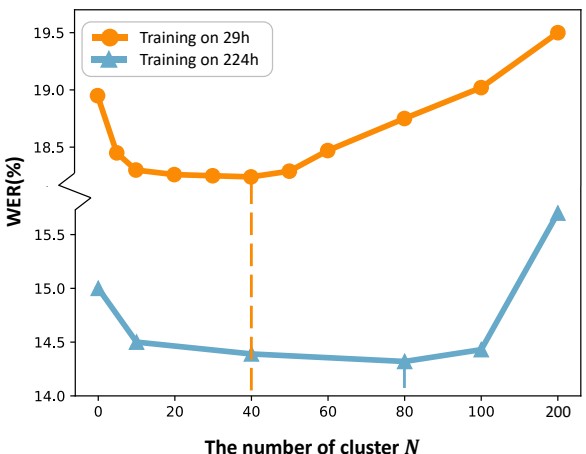

**Figure 5: Comparison of PCD$_{pro}$ performance with different number of clusters ($N$).**

*4.5.3* **The Impact of prompt length.** We conduct experiments to investigate the impact of prompt length on accuracy in Figure 5. Similar to the results with cluster center numbers, initially, as

the prompt length increases, it provides more information for the model, resulting in improved accuracy. However, beyond a certain point, the data becomes insufficient to train the corresponding parameters, leading to a decline in recognition capability. It is noteworthy that the inflection points obtained from the prompt method and the prefix method are different. This discrepancy arises because the prompt tuning method directly concatenates the prompt with the input, resulting in an increasing length of input at each layer. Consequently, excessively long prompts adversely affect the input.

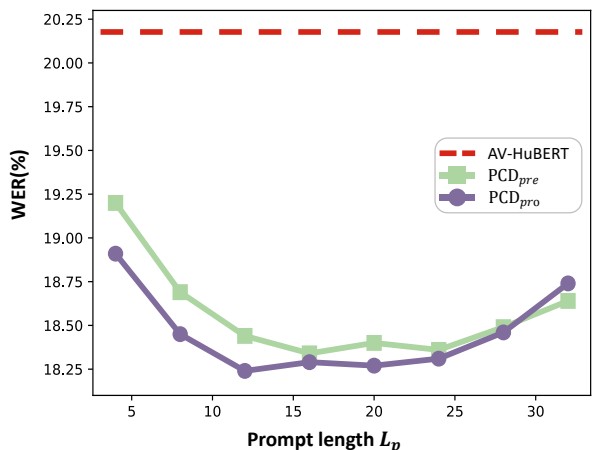

**Figure 6: Comparison of PCD performance with different prompt length ($L_p$).**

*4.5.4* **The Impact of Contrastive Decomposition.** In our proposed method, a contrastive decomposition framework is designed to constrain the generation of prompts. We investigate the impact of the framework as shown in Table 2. Compared to the scenario without using the framework (i.e., $\alpha = 0$), employing contrastive decomposition loss as a regularization term can effectively enhance the model's recognition accuracy. It enables the prompts to learn task-specific features and makes the model performance under modality-distortion closer to clean data. However, an excessive weighting of the contrastive loss may impede the model's learning of recognition capabilities. When the prompts themselves have not acquired sufficient knowledge, increasing constraints would be meaningless.

**Table 2: Parameter sensitivity of PCD$_{p}ro$ to different setting of contrastive decomposition framework.**

| Models | $\alpha$ | Wer (%) |
|---|---|---|
| AV-Hubert | - | 20.20 |
| PCD$_{pro}$ | 0 | 18.83 |
| | 0.001 | 18.78 |
| | 0.01 | **18.25** |
| | 0.1 | 20.01 |

*4.5.5* **Testing under extremely harsh conditions**. In handling the data with distortion, given the challenges in recognition and high distortion rates, we choose to discard scenarios where both modalities are distorted. However, this data may also be useful in practice. In this section, we investigate the performance of PCD in extremely harsh environments, where distortion can randomly occur in both modalities, as shown in Table 3. Apart from setting both modalities to have distortion to replace clean data, all other settings remain consistent, including adjusting the contrastive decomposition module to bring the specific-prompt for scenarios with both distorted modalities closer to those under single-modality distortion. As a result, PCD still brings improvements to model recognition accuracy. These improvements stem from the guidance on recognizing single-modal distorted data and the enhanced contextual understanding of recognizing multi-modal distorted data.

**Table 3: WER (%) performance of PCD in extreme harsh conditions. dis. represents distorted**

| Models | training set | $\eta$=90% | | $\eta$=70% | |
|---|---|---|---|---|---|
| | | $\mu$=20% | $\mu$=40% | $\mu$=20% | $\mu$=40% |
| AV-Hubert | 29h-clean | 21.13 | 35.24 | 25.04 | 39.43 |
| AV-Hubert | 29h-dis. | 14.20 | 21.76 | 16.73 | 25.49 |
| $PCD_{pre}$ | 29h-dis. | 13.40 | 20.12 | 15.14 | 24.54 |
| $PCD_{pro}$ | 29h-dis. | **13.23** | **19.13** | **15.01** | **23.07** |

*4.5.6* **Comparison with the missing modality.** Since distortion can impact model recognition, an obvious question arises: can we simply discard the modality-distortion to address the missing modality problem? Missing modality is another research hotspot in the multi-modal field, but similarly, there is no method that simultaneously addresses the presence of multiple missing conditions in the context of AVSR. We also evaluate PCD's performance with missing modality and compare it with distorted modality, presenting the results in Figure 7. When focusing on the missing scenarios in the left half of the image, we observe that PCD exhibits improvements across all levels of missing, particularly enhancing performance by 2.8% when the missing rate is at 100%. When considering the comparison between missing and distorted data across the entire figure, we observe that when samples are completely covered by distorted data, discarding the distorted modality entirely is a viable option. However, when the distortion covers only 80% or less of the samples, employing PCD leads to better results. The results demonstrate that the PCD method effectively guides the model not only in distortion settings but also in missing scenarios.

*4.5.7* **Enhancement for existing models.** Some outstanding AVSR approaches achieve high-precision recognition on clean data, yet this very attribute renders them highly susceptible to interference from fake segments. Since PCD involves adding an additional prompt module to a frozen model, we fine-tune the pre-trained AVHubert on the LRS3-distorted to improve the model's robustness to distortion while retaining its original performance, and the results are displayed in Table 4. Models trained on clean data exhibit poor performance when confronting modality-distortion.

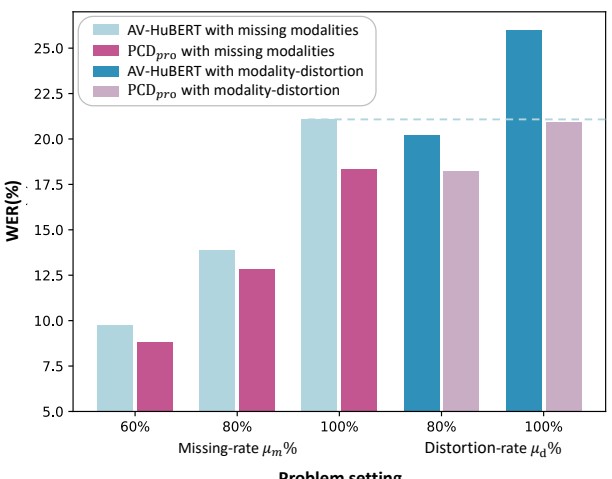

**Figure 7: The experimental results on missing modality and comparison with modality-distortion.**

Moreover, fine-tuning on distorted data notably impacts the original performance of the model on clean data. In contrast, models optimized with PCD, which only train a few parameters, greatly enhance the model's robustness to modality-distortion. Due to the considerable reduction in training data, there is a slight decrease in performance on clean data.

**Table 4: Comparison between existing models and PCD-optimized models on LRS3 dataset. dis. represents distorted**

| Models | training set | Param (MB) | $\eta = 70\%$ Wer(%) | | |
|---|---|---|---|---|---|
| | | | $\mu$=0% | $\mu$=60% | $\mu$=80% |
| AV-Hubert | 30h-clean | 161.5 | **4.08** | 29.32 | 39.04 |
| AV-Hubert | 30h-dis. | 161.5 | 6.58 | 18.23 | 20.89 |
| $PCD_{pre}$ | 30h-dis. | 3.84 | 4.10 | 17.13 | 19.22 |
| $PCD_{pro}$ | 30h-dis. | 3.84 | 4.13 | **16.74** | **18.83** |
| AV-Hubert | 433h-clean | 161.5 | **1.83** | 30.88 | 38.90 |
| AV-Hubert | 433h-dis. | 161.5 | 5.10 | 15.96 | 19.65 |
| $PCD_{pre}$ | 433h-dis. | 3.84 | 1.94 | 14.00 | 16.92 |
| $PCD_{pro}$ | 433h-dis. | 3.84 | 1.99 | **13.92** | **16.78** |

## 5 CONCLUSION

In this paper, we have proposed a novel method called PCD aiming to enhance robustness to modality-distortion in AVSR task. Concretely, we introduce prompt learning and design specific prompts for each type of modality-distortion to guide the model in adapting to the distortion. In order to effectively transfer knowledge from the high-quality domain obtained through pre-training to the low-quality domain with distortion, we employ two cluster-prompt strategies. In addition, to better fit task-specific features into prompts, we design a contrastive learning mechanism to constrain the generation of prompts. Extensive results on the newly-created benchmarks of modality-distortion speech recognition illustrates the superiority of our proposed method.

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
