# OpenReview forum: "Boosting Speech Recognition Robustness to Modality-Distortion with Contrast-Augmented Prompts"
_acmmm.org/ACMMM/2024/Conference — MM2024 Oral_

### Official Review · Reviewer_M2qD · 2024-05-12

**Rating:** 4
**Confidence:** 3

**Summary:**

This paper primarily tackles robustness challenges in audio-visual speech recognition tasks. To address these issues, the authors introduce a novel method called PCD (cluster-Prompt with Contrastive Decomposition). PCD leverages prompts to enhance the fine-tuning process of the model. Experimental results on the LRS2 dataset showcase the effectiveness of PCD, as it outperforms AV-HuBERT in both high-resource and low-resource scenarios, demonstrating superior performance across various conditions.

**Strengths:**

1. The paper offers a comprehensive examination of various speech and visual distortion scenarios, encompassing a diverse range of challenges that can arise in audio-visual speech recognition. This extensive analysis ensures a thorough understanding of the potential difficulties faced in this domain.
2. The authors present a novel prompt-based method and conduct meticulous ablation experiments to assess its effectiveness. The experimental design and analysis for the prompt-based approach are meticulously executed, providing a robust foundation for evaluating its impact on the overall system performance. This rigorous evaluation enhances the credibility and reliability of the proposed method.

**Limitations:**

1. The experimental evaluations in the paper focus solely on comparing their proposed method with AV-HuBERT and present results exclusively on the LRS2 dataset.
2. The introduction of the paper highlights the issue of audio-visual asynchrony. However, the paper does not delve into an in-depth analysis or propose specific techniques to address this problem.
3. They clarify that they are the first to enhance robustness in modality-distortion in speech recognition. But there are many papers focusing on this problem.

**Suitability:**

3

---

### Official Review · Reviewer_tCdk · 2024-05-22

**Rating:** 4
**Confidence:** 3

**Summary:**

This paper focuses on the modality-distortion problem in the audio-visual speech recognition task.  The paper introduces PCD (Cluster-Prompt with Contrastive Decomposition), a robust framework for Audio-Visual Speech Recognition (AVSR) that addresses challenges posed by modality-distortions like audio-visual asynchrony, video noise, and audio noise. PCD uses contrast-augmented prompts and cluster-based strategies to effectively transfer knowledge from high-resource to low-resource domains, improving speech recognition accuracy in various distorted scenarios. Experimental results on the LRS2 dataset demonstrate that PCD significantly outperforms existing methods, achieving state-of-the-art performance in handling modality-distorted data.

**Strengths:**

The key contribution of the paper is to leveraging the idea of AV-HuBERT on the distorted modalities. The cluster module is effective to relate the representations. Besides, the evaluation is extensive and shows great insight.

**Limitations:**

qualitative examples in the supplement materials do not show how the model can handle the distorted modalities effectively and how the clustering can bring the effect to the representation. It should present the visualization results on the cluster centers to demo how it performs a good prompt. For example, GroupVIT shows intermediate results. Besides, it should demo the spectrum of the audio to show how the proposed methods can effectively combat the distortion to extract the recognition features.
It mentions two parameters to indicate the percentage of the distortion but it does not mention the quantitive numbers such as SNR, SSIM. Besides, we need to keep distortion set up similar to real setup.

**Suitability:**

3

---

### Official Review · Reviewer_6svU · 2024-05-24

**Rating:** 5
**Confidence:** 3

**Summary:**

This paper aims to enhance the robustness of audio-visual speech recognition (AVSR) under modality-distortion conditions. It proposes two cluster-based strategies specifically designed for implementing the prompt mechanism. Additionally, a contrastive decomposition mechanism is introduced to constrain the explicit relationships among various modality conditions. Experimental results on the LRS2 dataset demonstrate that the proposed approach achieves state-of-the-art performance in audio-visual speech recognition under the constraints of distorted resources.

**Strengths:**

1. The authors propose a cluster strategy for prompt generation. For audio-visual features that exhibits greater similarity, cluster module facilitates prompts in offering more proximate guidance to the pre-trained model.
2. They design two types of prompt attachment strategies to integrate prompts with pre-trained models, transferring knowledge from a high-quality domain to a target low-quality domain.
3. They utilize a common-specific low-rank decomposition approach to supervise prompt learning for specific modal combinations with distortion. Contrastive learning is then applied to encourage similar tasks to learn more similar prompts, facilitating the extraction of task-related information and enhancing the effectiveness of guiding the model.
4. The paper conducts numerous experiments to rigorously validate the effectiveness of the method, and the experiments are detailed and methodical.

**Limitations:**

1. The extraction network mentioned in Section 3.3.1 consists of a linear projection layer, a summation operation, and a cluster-wise softmax layer. However, the summation operation is not depicted in Figure 2.
2. There are some minor discrepancies in the details presented in the paper, such as in Section 4.5.2, where it states, "The optimal performance under the 29h dataset is achieved with 30 cluster centers, whereas the number is 80 for the 224h dataset." According to the results in Figure 5, the optimal performance under the 29h dataset is achieved with 40 cluster centers.
3. In Section 4.5.3, "We conduct experiments to investigate the impact of prompt length on accuracy in Figure 5," should be changed to Figure 6.

**Suitability:**

2

---

### Meta-Review · Area_Chair_WDVP · 2024-07-03

**Recommendation:** Accept (Oral)
**Confidence:** 5

**Metareview:**

The paper addresses the problem of speech recognition. To this end, it proposes PCD: cluster-prompt with Contrastive Decomposition, a robust framework for modality-distortion speech recognition, specifically devised to transpose the pre-trained knowledge from the high-resource domain to the targeted domain by leveraging contrast-augmented prompts. In contrast to previous studies, it takes into consideration the possibility of various types of distortion in both the audio and visual modalities. The method is evaluated on the LRS2 dataset and it reports good performance in comparison to the state-of-the-art.

After considering the paper, the reviewer's comments, and the rebuttal I recommend 'accept' for the paper.

The reviewers highlight the following strengths and limitations:

Strengths:
1. The paper conducts numerous experiments to rigorously validate the effectiveness of the method
2. Leveraging the idea of AV-HuBERT on the distorted modalities, which makes the key contribution.
3. The authors present a novel prompt-based method and conduct meticulous ablation experiments to assess its effectiveness.

Limitations:
1. The introduction of the paper highlights the issue of audio-visual asynchrony. However, the paper does not delve into an in-depth analysis or propose specific techniques to address this problem.
2. The paper should clarify that the work is the first to address scenarios with multiple modality distortions,